# Functional anatomy and topographical organization of the frontotemporal arcuate fasciculus
Gianpaolo Antonio Basile[1], Victor Nozais[2,3], Angelo Quartarone[4], Andreina Giustiniani[4], Augusto Ielo[4], Antonio Cerasa [5], Demetrio Milardi[1], Majd Abdallah[6], Michel Thiebaut de Schotten [2,3], Stephanie J. Forkel [3,7,8,9] & Alberto Cacciola [1] ✉

Traditionally, the frontotemporal arcuate fasciculus (AF) is viewed as a single entity in anatomo-clinical models. However, it is unclear if distinct cortical origin and termination patterns within this bundle correspond to specific language functions. We use track-weighted dynamic functional connectivity, a hybrid imaging technique, to study the AF structure and function in two distinct datasets of healthy subjects. Here we show that the AF can be subdivided based on dynamic changes in functional connectivity at the streamline endpoints. An unsupervised parcellation algorithm reveals spatially segregated subunits, which are then functionally quantified through meta-analysis. This approach identifies three distinct clusters within the AF - ventral, middle, and dorsal frontotemporal AF - each linked to different frontal and temporal termination regions and likely involved in various language production and comprehension aspects. Our findings may have relevant implications for the understanding of the functional anatomy of the AF as well as its contribution to linguistic and non-linguistic functions.

The arcuate fasciculus (AF) is a prominent association pathway in the human brain. Since its first description in the 19th century, this white matter bundle has been considered crucial for language processing[1–4], and for other relevant cognitive functions[4]. Accordingly, this structure has been extensively investigated in the last decades, using post-mortem dissection methods and, more recently, in vivo diffusion-weighted tractography[5–8].

Anatomically, the AF has been subdivided into several segments: the long (frontotemporal or arcuate proper), anterior (frontoparietal, largely equivalent to the third branch of the superior longitudinal fasciculus, SLF3), and posterior (temporoparietal) segments[9,10]. Converging evidence from works combining tractography and functional MRI, or lesion mapping studies in clinical populations, suggests that these distinct anatomical segments may contribute at different levels in language-specific cognitive processes. The frontoparietal segment of the AF has been associated with phonology-to-movement mapping and phonology-based word retrieval[11,12], the parietotemporal segment to reading and word comprehension[13,14], and

the long segment to low-level phonemic and phonological processing in the context of language production, including word and non-words (sublexical) repetition[15,16]. While there is consensus that frontoparietal and parietotemporal segments represent distinct anatomical and functional units within the AF, the precise role of the direct, frontotemporal segment of the AF is still a matter of lively debate. A growing line of evidence suggests that the frontotemporal AF may take part in higher-order language production processes involving the integration of lexical and phonological information, such as naming and phonologic fluency[17–20]. It has been proposed that this functional dissociation between phonological and semantic processing in the frontotemporal AF may be grounded in the underlying bundle anatomy, as suggested by ex vivo and in vivo anatomical findings of "ventral" and "dorsal" frontotemporal sub-segments, with distinct course, origin, and termination[6,7]. It has been hypothesized that the ventral segment would mediate mostly phonemical-phonological processing, while the dorsal segment would be involved in lexical-semantical processes[21,22].

[1]Brain Mapping Lab, Department of Biomedical, Dental Sciences and Morphological and Functional Imaging, University of Messina, Messina, Italy. [2]Groupe d'Imagerie Neurofonctionnelle, Institut des Maladies Neurodégénératives-UMR 5293, CNRS, CEA, University of Bordeaux, Bordeaux, France. [3]Brain Connectivity and Behaviour Laboratory, Sorbonne Universities, Paris, France. [4]IRCCS Centro Neurolesi "Bonino Pulejo", Messina, Italy. [5]Institute of Bioimaging and Complex Biological Systems (IBSBC CNR), Milan, Italy. [6]Bordeaux Bioinformatics Center (CBiB), IBGC, CNRS, University of Bordeaux, Bordeaux, France. [7]Donders Institute for Brain Cognition Behaviour, Radboud University, Nijmegen, The Netherlands. [8]Max Planck Institute for Psycholinguistics, Nijmegen, The Netherlands. [9]Centre for Neuroimaging Sciences, Department of Neuroimaging, Institute of Psychiatry, Psychology and Neuroscience, King's College London, London, UK. ✉e-mail: alberto.cacciola0@gmail.com

However, potential functional dissociation within the frontotemporal arcuate has poorly been explored because of the lack of methods that conveniently combine tractography and functional MRI data.

Recently, novel methods advanced this framework by mapping the functional white matter networks identified by statistically combining task or task-free functional and structural imaging[23–25]. In addition, track-weighted dynamic functional connectivity (tw-dFC) is a recently developed technique that allows for a joint analysis of structural and functional connectivity (FC) by mapping time-windowed functional connectivity, sampled from resting-state functional MRI, back onto the underlying white matter anatomy, reconstructed by tractography[26]. In previous work, independent component analysis (ICA) applied to tw-dFC time series was suitable for identifying highly reliable and biologically meaningful functional units within the human white matter[27]. This feature makes it a valuable tool to parcellate white matter structures in an entirely data-driven fashion, based on the fluctuations of FC at their cortical endpoints.

In the present work, we adapted this method to characterize the functional anatomy of the human frontotemporal AF, building on the hypothesis that independent branches within the AF may be segregated by their distinct activity profiles. We obtained bundle-specific tw-dFC time series of the AF by combining high-quality resting-state and diffusion data[28,29]. Using a hard clustering approach based on ICA, we aimed at identifying anatomically and functionally dissociable AF clusters in an unsupervised, data-driven fashion, according to dynamic changes in FC at the streamline endpoints[29]. Finally, we peered into the functional meaning of such anatomical organization by applying a meta-analytic decoding approach based on the NeuroQuery predictive model and database[30].

## Results

### Functional activity in the arcuate fasciculus is best decomposed into two independent components

Preprocessed diffusion-weighted imaging (DWI) data of the primary, test-retest and validation datasets underwent an automatic AF reconstruction pipeline through TractSeg, an algorithm that directly segments white matter bundles from the Fiber Orientation Distribution (FOD) peaks[31]. For each voxel traversed by streamlines, tw-dFC time series at a given time window (~40-s length) were computed as the average FC value at the endpoints of the streamlines traversing that voxel (Fig. 1).

Bundle-specific tw-dFC volumes underwent a spatial group ICA framework implemented in the Group ICA of FMRI Toolbox (GIFT)[32,33]. Group analysis was performed separately for the primary, test-retest (Human Connectome Project, HCP) and validation (Leipzig Study for Mind-Body-Emotion Interactions, LEMON) datasets and for the left and right AF. For each cardinality of components ranging from $k = 2$ to $k = 5$, group ICA was successfully performed in all datasets for both the left and right AF. Measures

of between-subjects, within-subject, and between-cohorts spatial similarity and similarity to static FC (Functionnectome) were considered to select an optimal number of components for left and right AF parcellations (Fig. 2).

The similarity of group ICA results over split-half resamples of the main dataset was higher for $k = 2$ (both left and right $r = 0.99$), with slightly lower similarity for other k values. (Fig. 2A).

The within-subject similarity was found to be maximal for $k = 2$ both for left ($r = 0.98$) and right AF ($r = 0.98$); for left AF, lower values were obtained with $k = 3$, while for right AF, a drop in similarity values was observed for $k = 4$ and $k = 5$ (Fig. 2B).

The between-cohorts spatial similarity was substantially higher for $k = 2$ ICA solution both for left ($r = 0.90$) and right ($r = 0.89$) AF, with markedly lower values for higher values of k (Fig. 2C).

Finally, the similarity to functionnectome-based ICA was found higher for the $k = 2$ ICA solution both for left ($r = 0.79$) and right ($r = 0.85$) AF; the correlation was found to decrease gradually with increasing values of k, with a slight increase for $k = 5$ (Fig. 2D).

Taking these results together, $k = 2$ was selected as the optimal $k$ value for ICA analysis, and the resulting components were considered for AF parcellation.

### Independent components map on three anatomically distinct segments of the AF

The results of group ICA suggest a topographical organization of the left and right AF (Fig. 3A).

For the left and right AF, component maps included voxels of all the AF but with different weights, corresponding to distinct patterns of correlated and anti-correlated activity within the AF: for each voxel, positive weights indicate that the activity is positively correlated to the overall component time series. In contrast, negative weights indicate that the voxel activity negatively correlates with the component time series. This differentiation in voxel weights across the AF allows us to delineate the network's FC, highlighting the importance of understanding both synchronized and distinct patterns of neural activity. At the selected number of components of $k = 2$, the time series of these components, while not fully independent from each other, showed a relatively weak temporal correlation (left: $r = 0.17$, right: $r = 0.18$) (Fig. 3B).

Given the overlapping spatial distribution of these functional components, we opted for a hard parcellation of AF sub-units based on k-means clustering. Based on the silhouette plot (Fig. 4A), an optimal number of clusters of $c = 3$ was identified both for left and right AF.

Hard parcellation revealed a tripartite topographical organization of AF clusters following a ventral-dorsal topographical organization: a ventral cluster that extends from the superior temporal gyrus, superior temporal sulcus, and anterior part of the middle temporal gyrus to the most ventral portion of the inferior frontal gyrus; a middle cluster which connects the

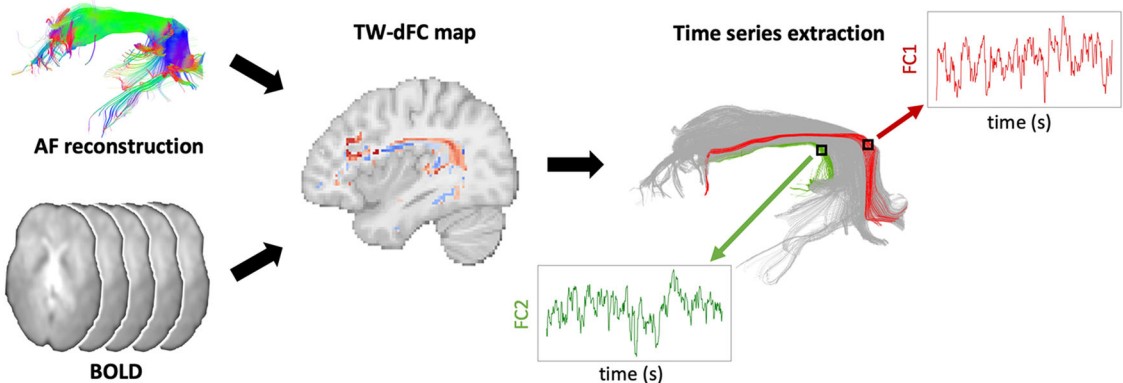

**Fig. 1 | Track-weighted dynamic functional connectivity (tw-dFC) of the arcuate fasciculus.** Data obtained from tractography and rs-fMRI are combined into a hybrid tw-dFC dataset. Preprocessed DWI data undergo an automatic AF reconstruction pipeline through TractSeg. For each voxel traversed by streamlines, tw-dFC time series at a given time window (~40-s length) are computed as the average functional connectivity (FC) value at the endpoints of the streamlines traversing that voxel.

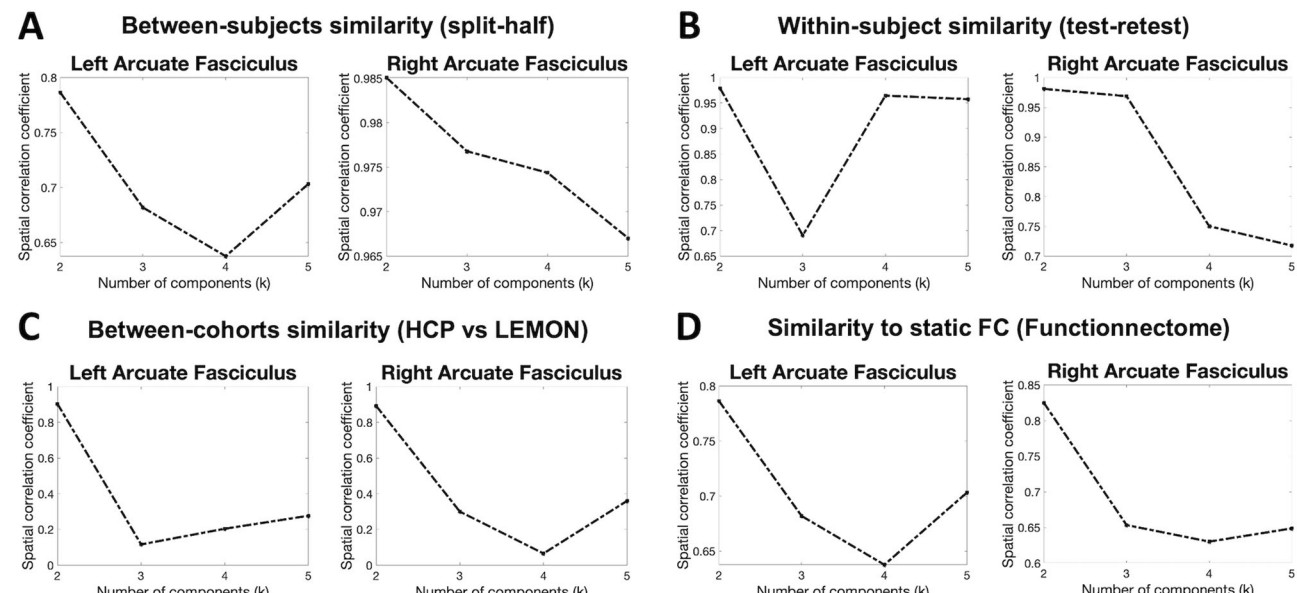

**Fig. 2 | Dimensionality selection measures. A** Between-subjects spatial similarity, calculated between symmetrical random halves of the main dataset; **B** Within-subject similarity, obtained from the test-retest dataset (**C**) Within-cohort similarity; **D** Similarity to static connectivity-based track-weighting results, calculated on the main dataset.

posterior middle temporal gyrus and anterior inferior temporal gyrus to the dorsal parts of the inferior frontal gyrus; and a dorsal cluster reaching the inferior temporal sulcus and middle frontal gyrus (Fig. 4B).

To clarify the relation between functional independent components (ICs), we plotted the average component weights from the group ICA for each of the three clusters (Fig. 4C). As highlighted from the plots, component 1 weights progress from extreme values in the ventral AF cluster to extreme opposite values in the middle cluster, while component 2 weights span from extreme values in the middle cluster to opposite extreme values in the dorsal cluster.

**Distinct AF clusters correlation patterns with meta-analytic maps**

A custom meta-analytic approach using the Neuroquery database was employed to functionally characterize white matter clusters from AF parcellation. Voxel-wise inverse distance maps were generated for each cluster centroid to create continuous distribution maps, indicating voxel proximity to centroids. These maps were masked with a mean GM mask and used in Neuroquery's image search to find top correlated neuroscience terms[30]. After removing duplicates and anatomical terms, unthresholded statistical maps for the remaining terms were converted to track-weighted predictive term maps using MRtrix3[34,35]. Pairwise Pearson's correlation quantified the similarity between cluster maps and term maps. Statistical significance was assessed with a permutational approach[36], and multiple comparisons were corrected using the Benjamini-Hochberg method. Correlation effect sizes were evaluated using the $R^2$ determination coefficient.

The analysis of correlation to track-weighted predictive term maps revealed dissociable correlation patterns for each AF cluster. Cluster inverse distance maps were correlated to meta-analytic terms, such that positive correlation values indicate that a meta-analytic map correlates to the proximity to cluster centroid (i.e., positively associated with a given cluster). Negative correlation values mean that a meta-analytic map is correlated to the distance from the cluster centroid (i.e., it is negatively associated with a given cluster). The initial screening of meta-analytic terms resulted in 33 neuroscience terms being selected for the following analysis (Supplementary Tables 1, 2). Supplementary Table 3 provides the links to the publications related to the meta-analytic terms.

In brief, the ventral AF cluster positively correlated with auditory-related terms ("pitch": left $r = 0.39$, $R^2 = 0.15$, $p < 0.001$; right $r = 0.41$, $R^2 = 0.17$, $p < 0.001$; "sound": left $r = 0.47$, $R^2 = 0.22$, $p < 0.001$; right $r = 0.51$, $R^2 = 0.26$, $p < 0.001$) and terms specific to voice processing ("vocal": left $r = 0.47$, $R^2 = 0.22$, $p < 0.001$; right $r = 0.57$, $R^2 = 0.33$, $p < 0.001$; "voice": left $r = 0.29$, $R^2 = 0.08$, $p < 0.001$; right $r = 0.50$, $R^2 = 0.26$, $p < 0.001$); it also

yielded positive correlation to phonology-related terms ("pseudo", which includes studies referring mostly to words-pseudowords discrimination: left $r = 0.40$, $R^2 = 0.16$, $p < 0.001$; right $r = 0.57$, $R^2 = 0.33$, $p < 0.001$; "phonological": left $r = 0.33$, $R^2 = 0.11$, $p < 0.001$; right $r = 0.55$, $R^2 = 0.30$, $p < 0.001$). It negatively correlated to terms related to semantic processing ("semantic": left $r = -0.19$, $R^2 = 0.03$, $p < 0.001$; right $r = 0.55$, $R^2 = 0.30$, $p < 0.001$; "semantic memory": left $r = 0.33$, $R^2 = 0.11$, $p < 0.001$; right $r = 0.55$, $R^2 = 0.30$, $p < 0.001$). Correlation values were generally higher for the right AF, and the right ventral clusters was also correlated to terms related to reading and text processing ("read": right $r = 0.56$, $R^2 = 0.31$, $p < 0.001$; left $r = 0.04$, $p > 0.05$).

The middle cluster of the AF positively correlated with most of the selected terms, with higher values for language-related terms, especially terms related to semantic processing ("semantic": left $r = 0.81$, $R^2 = 0.66$, $p < 0.001$; right $r = -0.24$, $R^2 = 0.06$, $p < 0.001$; "semantic memory": left $r = 0.77$, $R^2 = 0.59$, $p < 0.001$; right $r = -0.15$, $R^2 = 0.02$, $p < 0.001$, "semantic processing": left $r = 0.81$, $R^2 = 0.66$, $p < 0.001$; right $r = -0.03$, $p > 0.05$, "meaning": left $r = 0.79$, $R^2 = 0.63$, $p < 0.001$; right $r = 0.21$, $R^2 = 0.04$, $p < 0.001$, "noun": left $r = 0.75$, $R^2 = 0.56$, $p < 0.001$; right $r = 0.35$, $R^2 = 0.12$, $p < 0.001$) followed by syntactic ("syntactic": left $r = 0.56$, $R^2 = 0.32$, $p < 0.001$; right $r = 0.24$, $R^2 = 0.06$, $p < 0.001$, "syntax": left $r = 0.66$, $R^2 = 0.44$, $p < 0.001$; right $r = 0.44$, $R^2 = 0.19$, $p < 0.001$, "violations": left $r = 0.74$, $R^2 = 0.55$, $p < 0.001$; right $r = 0.67$, $R^2 = 0.44$, $p < 0.001$) and phonological processing ("phonological": left $r = 0.28$, $R^2 = 0.08$, $p < 0.001$; right $r = 0.14$, $R^2 = 0.02$, $p < 0.001$). It also yielded correlations to non-linguistic terms such as those related to higher-order cognitive functions ("consideration": left $r = 0.46$, $R^2 = 0.21$, $p < 0.001$; right $r = 0.41$, $R^2 = 0.17$, $p < 0.001$, "campus" which mostly refers to works on autobiographical memory: left $r = 0.40$, $p < 0.001$; right $r = 0.39$, $p < 0.001$) and social function ("social cognition": left $r = 0.68$, $p < 0.001$; right $r = 0.62$, $p < 0.001$). Overall, correlations with language-related terms were weaker or negative in the right hemisphere, while correlations to social-related terms were slightly stronger. The middle cluster distance map also negatively correlated with purely auditory-related terms.

Finally, the dorsal cluster of the AF was anti-correlated with terms related to acoustic or language processing. It shared with the middle cluster positive correlation to terms related to higher-order cognitive functions ("campus", which includes studies related to autobiographical memory processes: left $r = 0.33$, $R^2 = 0.11$, $p < 0.001$; right $r = 0.42$, $R^2 = 0.18$, $p < 0.001$; "consideration": left $r = 0.39$, $R^2 = 0.15$, $p < 0.001$; right $r = 0.41$,

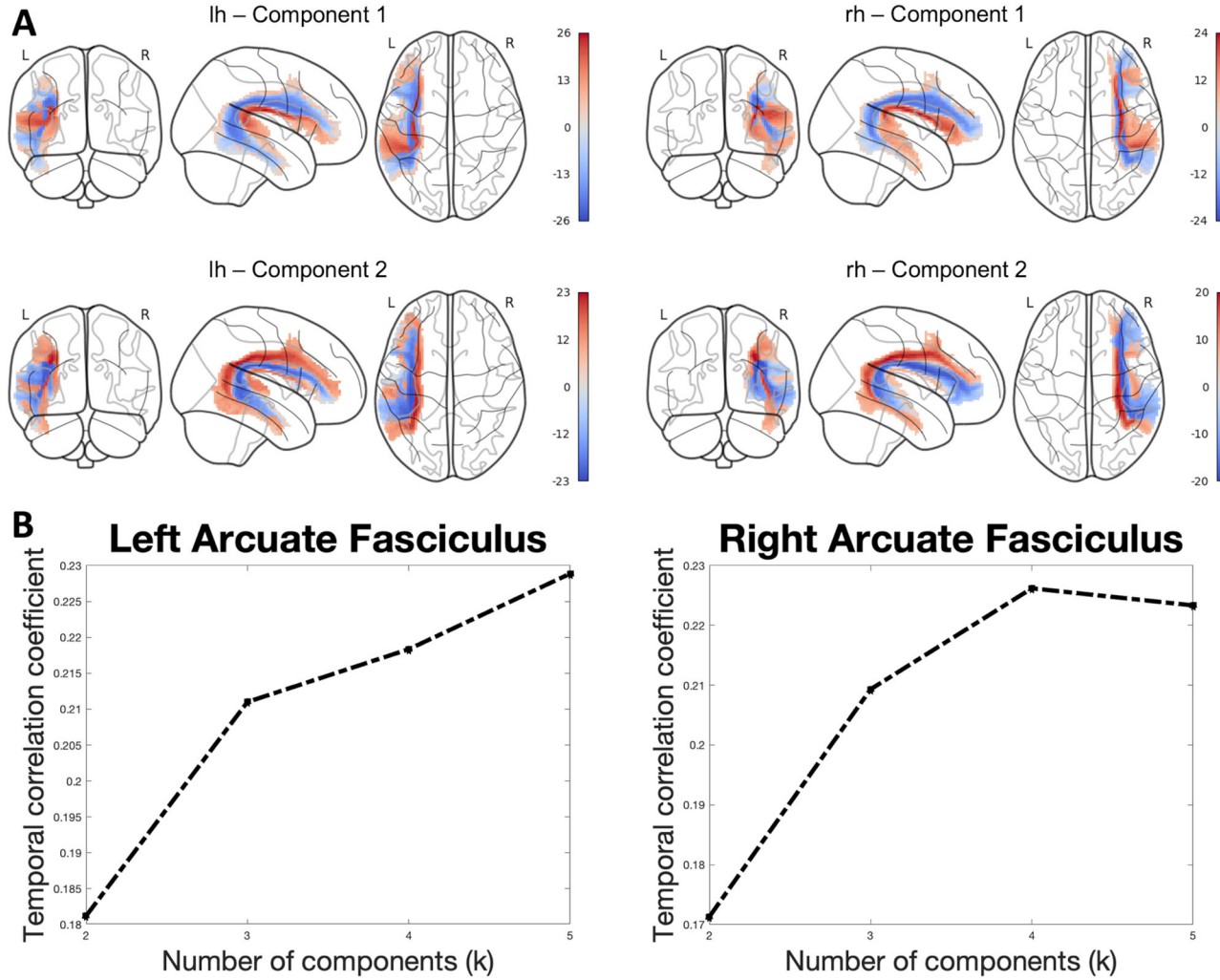

**Fig. 3 | Independent components as modes of dynamic brain activity along the frontotemporal AF. A** Spatial maps of the independent components for left and right AF. Group-level component z-maps are obtained from back-reconstructed components in all participants of the main HCP dataset. Since components from ICA have intrinsic sign indeterminacy, right component 2 has been sign-flipped to emphasize the similarity to its left counterpart. **B** Average pairwise correlations between component time series. At the given value of $k = 2$, temporal independence across components is higher (lower correlation between time series). lh left hemisphere, rh right hemisphere.

$R^2 = 0.16$, $p < 0.001$ "locked": left $r = 0.50$, $R^2 = 0.25$, $p < 0.001$; right $r = 0.55$, $R^2 = 0.18$, $p < 0.001$) and to semantic memory in the right hemisphere (right $r = 0.39$, $R^2 = 0.16$, $p < 0.001$; left $r = −0.02$, $p > 0.05$). The most relevant terms for each cluster are summarized in word clouds (Fig. 4D).

## Discussion

We fractionated the anatomy of the frontotemporal AF in the healthy adult human brain in the left and right hemispheres based on resting-state fMRI differences and characterized functionally these subcomponents through comparison with metanalytic maps. Three main findings emerged from our work. First, distinct patterns of correlated and anti-correlated activity exist within the AF. Second, the frontotemporal arcuate can be subdivided into three anatomically distinct clusters. Third, these divisions connect areas classically activated by different fMRI paradigms.

### Structure and function in the AF: independent components of dynamic brain activity along the AF

The present work provides an account of the functional anatomy of the AF in the human brain, by investigating tract-specific, time-dependent fluctuations in spontaneous functional activity. This paradigm has been recently employed to identify spatially independent functional units in the whole-brain white matter, representing fiber bundles sharing coordinated

oscillations of connectivity at their endpoints[27]. Herein, we leveraged this method to elucidate the structure-function relationship in the AF.

By applying ICA to the track-weighted, time-varying FC data, we identified distinct patterns of correlated and anti-correlated activity within the frontotemporal AF. In particular, we described that the patterns of functional activity within the AF are best characterized by the least possible number of ICs ($k = 2$), as confirmed by between-subject and within-subject (test-retest) reliability measures. Additionally, the two components identified by group ICA showed very high out-of-sample reproducibility (above 0.90), suggesting that they may capture functional features of AF that are robust to experimental differences in data acquisition and processing. It is worth noting that the two datasets employed in the analysis showed several demographical (larger age range, different gender proportion) and technical differences both in DWI and rs-fMRI acquisition and processing[29,37–39] further highlighting the robustness of these results.

While maintaining a well-recognizable polarity between different clusters of the AF, the activity patterns highlighted by group-ICA components are not entirely segregated to previously anatomically defined segments of the arcuate, but span across the whole tract. In other words, the spatial patterns highlighted by ICA may be interpreted as spatially distinct and temporally coherent "modes" of dynamic connectivity along the AF, in

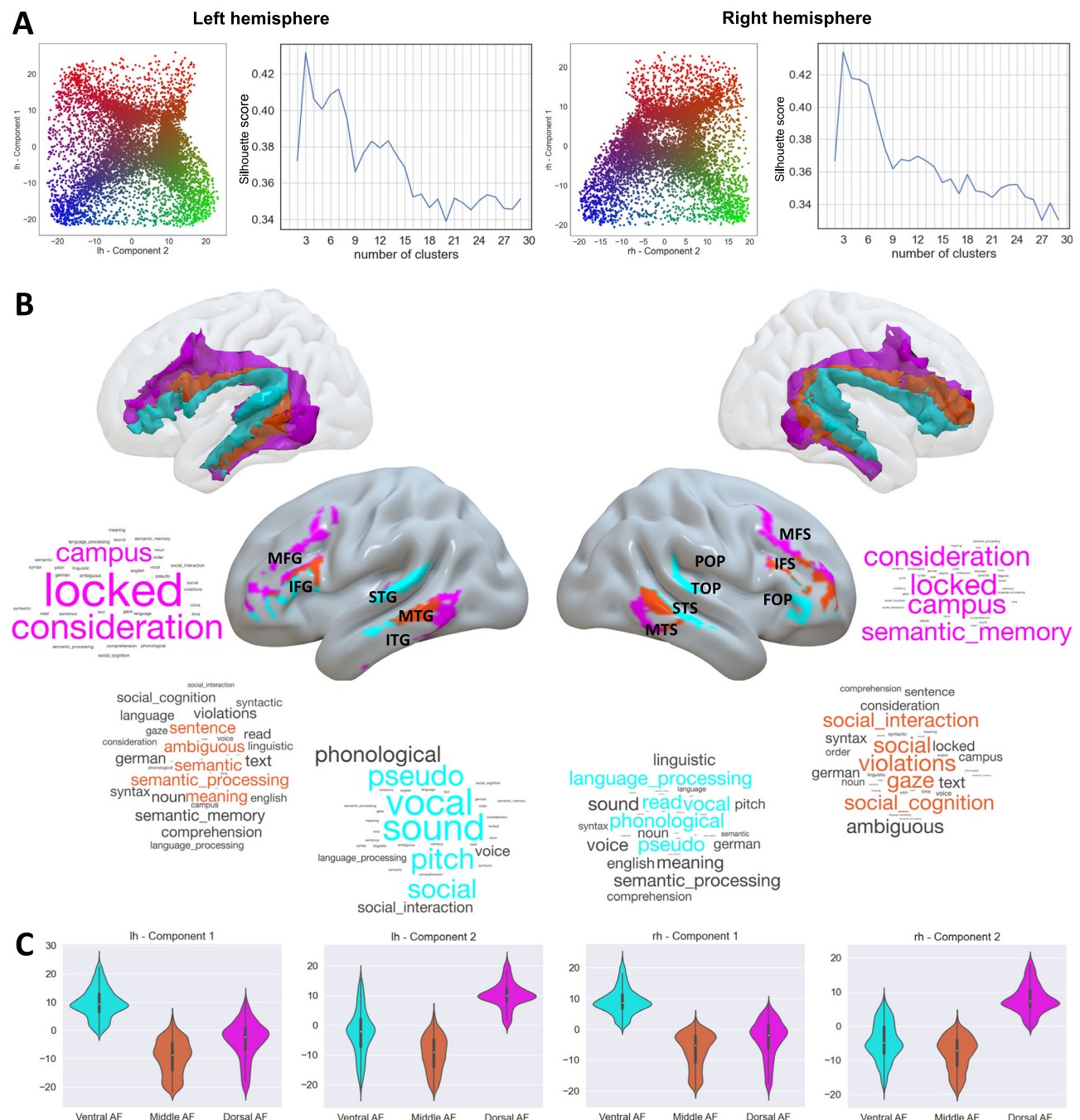

**Fig. 4 | Clustering of component spatial maps reveals a tripartite organization of the AF. A** Scatter plots of component values in the component space and the resulting silhouette plots. For each voxel, weights of component 1 and component 2 are plotted in the resulting component space; colors progress from blue to red across component 1 and from blue to green across component 2. **B** Ventral, middle, and dorsal clusters of the AF and their meta-analytic characterization. Clusters are both rendered in 3D volumetric space and projected on the cortical surface to emphasize the cortical termination patterns. Labels indicate gyral structures on the left and

sulcal structures on the right. MFG middle frontal gyrus, IFG inferior frontal gyrus, STG superior temporal gyrus, MTG middle temporal gyrus, ITG inferior temporal gyrus, MFS middle frontal sulcus, IFS inferior frontal sulcus, FOP frontal operculum, POP parietal operculum, TOP temporal operculum, STS superior temporal sulcus, MTS middle temporal sulcus. Word clouds are derived from correlation values obtained from meta-analytic decoding. Cyan: ventral cluster, orange: middle cluster, magenta: dorsal cluster. **C** Violin plots of component weights across the three clusters. lh left hemisphere, rh right hemisphere.

which distinct portions of the tract show anti-correlated dynamic connectivity (i.e., while FC increases at the endpoints of a certain cluster within the tract, it decreases at the endpoints of another cluster and vice-versa).

In recent years, a growing interest in FC's dynamic, time-varying properties has arisen[40]. While the exact biological meaning of such fluctuations is still far from being fully understood[41], there is substantial consensus that they may reflect, to a certain extent, neuronal sources of activity at different

frequency bands[42–44]. In addition, emerging evidence suggests that time-varying FC is partly constrained by anatomical connectivity[43,45] and that it may be related to spontaneous mental processes, such as arousal, perceptual fluctuations, mind-wandering, or daydreaming[46–48]. In keeping with this view, projecting the spatial patterns of fluctuations in FC at the endpoints of the AF onto the tract itself may help understand how different portions of the tracts constrain intrinsic activity and whether they show dissociable functional profiles during spontaneous mentation.

Finally, FC fluctuations at rest have been proposed to reflect interactions between functional systems at the network level during spontaneous activity ("brain states")[49] or during tasks[50]. These dynamic rearrangements of functional networks, including context-dependent modulation of within- and between- network connectivity ("meta-networking") have been hypothesized to be relevant to complex brain functions, including language[51]. In this framework, the pattern of FC fluctuations observed at the endpoints of distinct segments of the AF may result from the dynamic recruitment of functional modules involved in linguistic processes during mental activity.

To gain additional insight into the functional meaning of these ICs, we benchmarked our results against those obtained with a similar method, the Functionnectome[24]. While not entirely identical to tw-dFC, this method also involves the mapping of functional data on white matter priors to combine structural and functional information, and can be applied to resting-state data[25]; on the other hand, as it is based on direct mapping of BOLD signal onto the white matter, the resulting components can be seen as an estimate of "static" FC units in the AF, compared to the time-windowed dynamic FC of tw-dFC.

The high correlation observed between the components identified with these methods (Fig. 2) is in line with previous accounts describing relatively similar results between "static" and dynamic connectivity-based ICA of resting state data[27,52]. This suggests that fluctuations in FC follow the same spatial patterns of functional activity and are organized along the same white matter pathways.

## Three new divisions of the frontotemporal arcuate fasciculus

We applied a clustering algorithm to the component weights to explore the inherent anatomy underlying the functional patterns revealed by spatial ICA. We observed that the spatial distribution of components points out a tripartite subdivision of the AF with a defined ventro-dorsal and latero-medial topography.

A similar subdivision of fiber bundles in the frontotemporal AF has been described by various studies, both in vivo using diffusion tractography. While the most widely accepted model of AF anatomy substantially regarded the frontotemporal segment as a whole[9], a substantial body of anatomical evidence suggests that it may be instead composed of multiple, potentially independent units. Anatomical findings based on in vivo tractography and ex vivo fiber dissection subdivided the frontotemporal AF into an inner or ventral pathway, interconnecting the pars opercularis and the most ventral portion of precentral gyrus with the STG and rostral MTG, and an outer or dorsal pathway that connects ventral precentral gyrus, caudal middle frontal gyrus, and dorsal pars triangularis/dorsal prefrontal cortex with MTG and ITG[6,7]. However, it is worth noting that anatomical methods alone are inherently blind both to the exact origin and termination of fiber bundles and that the functional segregation of these distinct segments can only be assumed as a hypothesis. Our data-driven subdivision is in line with the overall ventral-dorsal topography identified by these investigations by showing a ventral cluster running between the posterior STG and anterior MTG and the ventral pars orbitalis and triangularis, as in ref. 6, and two outer-most clusters (middle and dorsal) corresponding substantially to the anatomically-defined dorsal AF: the middle cluster connects the posterior MTG and ventral frontal lobe, while the dorsal cluster connects the dorsal pars triangularis, middle frontal gyrus, and ventral premotor cortex to the posterior MTG and ITG.

Lastly, our results suggest a slightly asymmetrical pattern between the left and right AF sub-units. However, in contrast to the available literature[7], we did not observe marked left-right differences in the terminations of the ventral cluster; rather, different termination patterns were demonstrated in the middle cluster, which in left hemisphere terminates mostly on the dorsal pars triangularis, while in the right hemisphere expands more to the middle frontal gyrus (that in the left hemisphere is covered mostly by the dorsal cluster).

## Functional characterization of AF clusters: relevance to language processing and cognitive function

Along with providing a multimodal, FC-informed, and data-driven anatomical segmentation of the AF, we also sought to elucidate the cognitive relevance of our model by applying a custom-designed meta-analytic decoding paradigm to the AF clusters derived from ICA-based parcellation.

Since earlier investigations, anatomical models of AF and its structural subunits have been tightly linked to functional language processing models[9]. There has been a considerable effort towards clarifying the role of AF in the context of the so-called "dual stream model" of information flow between language-related areas, which is anchored to specific white matter tracts[53,54].

This model has been formulated in terms of two diverging, parallel but mutually interacting cortical streams related to speech production: a ventral stream involved in mapping sound to meaning (semantic processing) and a dorsal stream involved in mapping sound to motion (phonological processing)[55,56]. Within this model, there is general agreement that the AF is primarily involved in dorsal stream functions. At the same time, other white matter structures (e.g., uncinate fasciculus, inferior fronto-occipital fasciculus (IFOF)) have been proposed to subserve ventral stream functions[57–60]. However, whether the AF may be involved in ventral stream functions, such as lexical or semantic retrieval, is still a matter of lively debate[17–20].

Our meta-analytic decoding found partially overlapping yet dissociable correlation profiles between our AF clusters and language-related terms. The ventral cluster of AF, covering the superior and anterior middle temporal gyrus, was correlated with the results of functional imaging studies concerning pitch and voice recognition. Evidence from direct electrical stimulation (DES) and microelectrode recording in awake surgical patients suggests that the STG may be involved in syllable and word recognition[61–64] and pitch[65]. In keeping with these findings, DES in the middle and superior temporal gyrus and peri-insular white matter results in word deafness and phonemic paraphasia[18,66–68], strengthening the hypothesis that the ventral AF may be involved in sensory and motor phonological processes.

Conversely, the middle cluster of the AF, connecting the dorsal inferior frontal gyrus and the posterior middle temporal gyrus and sulcus, correlated with semantic processing. This finding is in line with the involvement of these regions in semantic tasks, as suggested by non-invasive stimulation and lesion studies[69–71]. Semantic access and retrieval have been described to be mediated by other white matter structures, such as the IFOF[20,71,72]. However, a recent observational study on patients with unilateral left hemisphere stroke suggests that structural connectivity estimates in the dorsal AF may be related to semantic functions[73]. Similarly, a recent tractography and task-based fMRI investigation reported that structural integrity of the left AF segment connecting the middle temporal gyrus with the dorsal pars opercularis predicts performance in a lexical-semantic verb-generation task[22], suggesting that this portion of AF may be partially involved in semantic processing, especially during speech production. Our middle AF cluster is closely similar to the dorsal, "semantic" AF subtract identified by Janssen et al.[22]. This AF cluster was mostly, but not exclusively, correlated to semantic processing-related terms. At the same time, it showed a significant correlation with almost all the language-related terms, suggesting that this part of the AF may have a role in the integration between ventral and dorsal stream processing.

Lastly, the most dorsal cluster of the AF, connecting ITG to the middle frontal gyrus, strongly anticorrelated with phonological and semantic language terms, in partial contrast with studies suggesting a putative linguistic role of the inferior temporal terminations of the AF[19,74–76].

Along with enforcing the notion of different functional implications for different segments of the left AF, our findings also shed light on the functional significance of the right AF, which is far less understood than its left hemisphere homolog[77,78]. Our results suggest that the right and the left AF share a substantially similar morpho-functional organization, although with some relevant differences. The ventral cluster of right AF, similarly to its left homolog, shows a high correlation with phonological functions, voice, and pitch recognition; for some terms, the correlation was even higher than for the left AF (e.g., "voice", "phonological", "read"). Traditionally, "core" functions of language processing such as phonological, lexical, and semantic processing are considered as left-lateralized. In contrast, the right hemisphere is thought to be involved in processing "secondary" language functions such as prosody, pitch, and intonation[79–82]. It is then possible that the

correlation pattern found for the right hemisphere may reflect non-specific language-related activity in the contralateral hemisphere in task-based studies focused on phonological and semantic features of language processing in general. However, recent evidence from post-stroke patients suggests that the right AF may play a role in the long-term recovery of language functions, likely compensating for the loss of function of the contralateral AF[83,84]. Analogously, compensatory strategies for language functions in the right AF have been described in patients with psychosis[85]. This may suggest that, at least partially, the right AF may be involved in primary language functions such as phonological or semantic processing, even in a healthy brain. In keeping with its role in mediating the emotional components of language processing, the right AF has also been suggested to be relevant for social cognition. Some studies described a relation between social cognitive functions, such as emotional intelligence, mentalization, or mind-reading abilities, and AF integrity and microstructure[86–88]. Our study observed a higher correlation between social cognition and social communication for the middle cluster of the AF. Of note, this component also shows the most marked asymmetry in terms of frontal termination sites between the left and right hemispheres, which may concur to explain why correlation to social cognition terms is higher in the right hemisphere. In this purely explorative context, the identified finding allows us to hypothesize a functional dissociation between social and linguistic processing within the right AF. This parallels the phonological-semantical dissociation previously proposed for the left AF. These results provide valuable insights into the anatomical substrates of various linguistic and non-linguistic functions dependent on AF integrity and connectivity. In synthesis, while comparing the results of our functional and anatomical parcellation of the AF at rest to meta-analytic estimates derived from task-based functional MRI does not provide conclusive evidence on the involvement of specific tract segments, our work allows to draw hypotheses on the functional meaning of specific anatomical segments of the AF. The confirmation of these findings is therefore left to further investigations, through intra-operative electrode recording during awake surgery[18].

### Technical issues and limitations

Some limitations of the current approach need to be acknowledged. First, the validation dataset does not perfectly match the primary dataset, showing relevant differences in several experimental conditions. Such differences include demographical (larger age range, different gender proportion) as well as technical factors affecting both DWI (single shell, low b-value, no filtering) and rs-fMRI (lower temporal resolution, different denoising pipeline). While we chose to maintain the differences between datasets to highlight the reliability of our findings further, this approach makes it impossible to quantify the extent to which differences observed between the two datasets are driven by differences in the experimental pipeline rather than genuine, inter-individual anatomical and functional variability.

Second, we adopted an automatic tract reconstruction algorithm based on machine learning[31] paired with deterministic tractography, which is known to provide more conservative estimates of white matter trajectories[89]. As such, we adopt the definition of the AF as characterized in the TractSeg software, which in turn adopts a tract definition system based on cortical termination patterns[90]. Equally valid, automatic segmentation algorithms for AF reconstruction[91,92] may provide slightly different anatomical reconstructions, potentially affecting the results.

Third, choosing a pre-defined anatomical scaffold of the AF to guide tw-dFC generation forces the assumption that FC fluctuations sampled at the bundle termination are entirely driven by activity along the AF. This excludes the contributions of other fiber tracts that may share the same cortical projections, such as the uncinate fasciculus[93], superior longitudinal fasciculus[94], IFOF[95], or short-range U-fibers. This consideration also extends to non-neuronal sources of FC fluctuations[96].

Fourth, the quality of the tw-dFC maps strongly depends on the correct alignment of tractography and BOLD-fMRI data, making non-linear registration of tractograms, and distortion correction of both DWI and BOLD data critical steps for accurate and reliable results.

Lastly, meta-analytic decoding provides a quantitative estimate of similarity between a given brain map and meta-analytic maps derived by collating results from multiple imaging studies[97]. The NeuroQuery approach synthesizes meta-analytic maps for each term from studies in which that term, or other semantically related terms, are frequently mentioned[30]. This approach was preferred over other meta-analytic approaches as it permits obtaining meaningful statistical activation maps even for terms with few studies available, unlike other methods that are based on a reduced set of cognitive latent variables ("topics")[98]. However, the resulting statistical maps do not correspond to any specific fMRI task or mental state and do not necessarily represent a single task-positive functional network[99].

## Conclusion

Our investigation demonstrates that the AF may be subdivided into two ICs according to dynamic changes in FC at its streamline endpoints. These ICs spatially subdivide AF into three clusters with distinct courses and cortical termination. Findings from our custom meta-analytic approach suggest that each cluster may be related to distinct functional processes. Our results support a partial dissociation between auditory/phonemic and lexical/semantic functions in ventral vs middle left AF, possibly paralleled by a partial dissociation between auditory-phonemic and social communication processing in ventral vs middle right AF. At the same time, the dorsal cluster is involved in non-linguistic processing in both hemispheres. Our findings provide data-driven evidence for the morpho-functional segregation of the frontotemporal AF in both hemispheres.

## Methods

### Data acquisition and preprocessing

All the analysis relevant to the present work has been performed on two separate datasets: a main dataset and a validation dataset. The main dataset consists of 3T structural, diffusion, and resting-state functional MRI data were obtained from the HCP repository (https://humanconnectome.org). Specifically, two distinct subsamples of patients from the HCP collection have been employed for the present work: the first subsample (*primary dataset*) consisted of 210 healthy participants (males = 92, females = 118, age range 22–36 years), and the second (*test-retest dataset*) included 44 participants with available test-retest MRI scans (males = 13; females = 31; age range: 22–36 years)[38]. The validation dataset includes 3T structural, diffusion, and rs-fMRI data of 213 healthy subjects (males = 138, females = 75, age range 20–70 years) from the LEMON dataset (https://www.nitrc.org/projects/mpilmbb)[29].

Left-handed participants have not been excluded from the analysis as representing less than 10% in both samples (9.04% in the HCP dataset; 9.81% in the LEMON dataset).

The two datasets come with remarkable differences in demographics, data acquisition and preprocessing, that are described in full details in Supplementary Methods, as well as in the relevant reference publications[29,37–39]. Notably, it was not possible to statistically assess differences in age due to the differing age ranges between the two datasets. However, the first dataset (HCP-young adults) consists exclusively of young adults aged 25–35, while the second dataset combines a subsample of young adults (aged 20–35) with a subsample of healthy older adults (aged 59–77). Gender differences are significant across the two datasets as the LEMON dataset has a significantly higher proportion of female participants ($\chi^2 = 18.14$, $p < 0.001$). The DWI scans of the two datasets underwent different preprocessing pipelines: namely, the HCP data (multi-shell, 90 directions per shell at $b = 1000$, $b = 2000$, $b = 3000$, 1.25 mm³ voxel size) were available in a minimally preprocessed form, while the LEMON DWI scans (single shell, $b = 1000$, 64 directions, 2 mm³ voxel size) were available only in raw form and were preprocessed entirely with a dedicated pipeline using the MRtrix3 software[100]. Both HCP (2 mm³ voxel size, TR = 720 ms) and LEMON (2 mm³ voxel size, TR = 1400 ms) rs-fMRI data were obtained in standard-space (MNI152, 2 mm³) preprocessed and denoised form, though the featured preprocessing steps are different between the two datasets.

For the LEMON dataset, we employed structural T1-weighted scans to calculate direct and inverse transformations to standard MNI152 space, while for the HCP dataset, they were already made available as part of the preprocessed dataset.

Of note, we decided to keep the preprocessing pipelines different, as in our previous work[27], considering the different acquisition features of the two datasets and in order to further highlight the reproducibility of our findings.

### Bundle-specific tractography and tw-dFC

Diffusion signal modeling was performed on the preprocessed DWI data using the constrained spherical deconvolution (CSD) framework, which estimates white matter FOD function from the diffusion-weighted deconvolution signal using a single fiber response function as reference.[101]. For HCP DWI data (multi-shell), a multi-shell multi-tissue (MSMT) CSD signal modeling algorithm was applied to estimate separate response functions in WM, GM, and CSF[102]. For LEMON DWI data (single-shell), a single-shell 3-tissue (SS3T) CSD signal modeling was applied, despite the very low b-value, to keep the processing as consistent as possible between the two datasets, since it is necessary for the following automatic tract extraction step. In addition, it has been suggested to outperform the tensor model for tract reconstruction even at very low b values[103]. SS3T-CSD is a variant of the MSMT model optimized for RF estimation in single-shell datasets and was performed using MRtrix3Tissue[104], a fork of MRtrix3 software (https://3tissue.github.io).

A robust and unbiased reconstruction of the AF is of key importance for the good quality and generalizability of results. For bundle-specific tractography of the frontotemporal AF, TractSeg (https://github.com/MIC-DKFZ/TractSeg/), a convolutional neural network-based tract segmentation approach, was employed. The TractSeg algorithm directly segments white matter bundles for each subject from the FOD peaks; importantly, this method has been demonstrated to be less affected by the original data quality compared to other automatic tract segmentation algorithms[31], and was then preferred to grant high comparability of the results between the different data-quality HCP and LEMON datasets.

Participant-level binary tract masks and ending masks obtained from TractSeg were employed to guide tractography of the frontotemporal AF, which was performed on FODs using deterministic tractography (SD-STREAM algorithm)[105] with default tracking parameters up to a fixed number of 2000 streamlines.

For each participant of both HCP and LEMON datasets, tractograms of left and right AF were registered to the MNI 152 standard space by applying the aforementioned non-linear transformations, to bring individual tractography and rsfMRI data in the same space for tw-dFC analysis. Then, tractograms were combined with preprocessed rs-fMRI time series using MRtrx3's tckdfc command to generate a 4-dimensional tw-dFC time series[26]. Separate tw-dFC time series were obtained for the left and right frontotemporal AF, and the following parameters were used: spatial resolution: 2 mm2, sliding window shape: rectangular, sliding window length: ~40 s (55 time points for the HCP data, TR = 0.72 s; 29 time points for the LEMON data, TR = 1.4 s). The length of the sliding window was chosen in line with previous works to maximize the stability of the time-varying connectivity profiles[52,96,106,107]. For the HCP data, tw-dFC derived from LR and RL phase encoding volumes were temporally concatenated for each participant. Estimation of tw-dFC of the AF is summarized in Fig. 1.

### Group-level analysis

**Group ICA-based parcellation.** Bundle-specific tw-dFC volumes underwent a spatial group ICA framework implemented in the GIFT[32,33]. Group analysis was performed separately for the primary dataset (HCP) and the validation dataset (LEMON) and for the left and right AF. For each AF tract, a binary mask for group ICA was built after transforming all the subject bundle masks to template space by summing up all individual masks and applying a probability threshold of 25% (i.e., voxels that were part of the AF in at least 75% of participants were considered for group analysis). The pipeline for group ICA analysis involved (1) a first

dimensionality reduction step in which a subject-level principal component analysis (PCA) was applied to tw-dFC data to obtain 50 principal components per subject; (2) a second step in which dimensionality-reduced data of all subjects were temporally concatenated and a secondary PCA dimensionality reduction was applied along directions of maximal group variability; (3) the proper group ICA step, in which a given number (k) of ICs is obtained from low-dimensional data using the Infomax algorithm[108]; (4) back reconstruction, to obtain subject-specific spatial maps and time courses for each components using the group information guided ICA (GIG-ICA) algorithm[109]; (5) back-reconstructed individual components were then averaged and normalized to obtain group-level z-maps of each component.

**Data-driven dimensionality selection.** To select the most appropriate number of components (k) for AF parcellation given our data, the ICA pipeline was iterated for the left and right AF separately at different values of k, ranging from 2 to 5, and for each ICA solution and the following measures were calculated:

1. Between-subjects spatial similarity: the primary dataset (HCP, 210 subjects) was split into symmetrical, random halves (105 subjects each) and the group ICA pipeline was run on each split for all the k values; the average Pearson's correlation coefficient between group spatial ICA maps of the first and second split was employed as a measure of split-half similarity;
2. Within-subject spatial similarity: the test-retest dataset (HCP, 44 subjects with test-retest diffusion and rsfMRI data) was employed. The group ICA pipeline was run separately on test and retest tw-dFC data for all the k values; the average Pearson's correlation coefficient between group spatial ICA maps of the test and retest data was employed as a measure of test-retest similarity;
3. Within-cohorts spatial similarity: the primary dataset (HCP, 210 participant) and the validation dataset (LEMON, 213 participants) were considered. Group ICA was performed separately on each dataset for all the k values; the average Pearson's correlation coefficient between group spatial ICA maps of the primary dataset and the validation dataset was employed as a measure of external validation.
4. Similarity to "static" FC-based ICA: we benchmarked our results against a recently developed algorithm that involves mapping of the function signal from fMRI to tractography-derived priors of white matter anatomy, the "Functionnectome"[24]. In a recent implementation, this method has been extended to resting-state FC[25] Here, we employed the participant-level tractograms of left and right AF to derive group-level AF tractography priors. Then, we applied the Functionnectome algorithm to map the functional signal from individual BOLD volumes on these priors. The resulting 4-dimensional volumes underwent group ICA with the same parameters as for tw-dFC data. The group-ICA results for each value of k were compared using the average Pearson's correlation coefficient.
5. Functional correlation between component time series: to investigate the temporal independence of time series at each value of k, we employed the back-reconstructed time series for all subjects of the main dataset. Pearson's correlation was calculated pairwise for each pair of components on each individual and then averaged across the entire dataset. For values of k > 2, the average between pairwise correlation values was considered.

The optimal number of components was decided based on a consensus approach between all these metrics: the value k for which most metrics showed the highest value.

**Component-driven AF parcellation.** To derive a parcellation of the AF from the component maps, we applied a k-means clustering procedure in the component space. Briefly, after selecting an optimal number of components according to the measures described above, each voxel in the AF was clustered according to its similarity in weight in each component

z-map. The ideal number of clusters (c) was determined using the silhouette coefficient[110].

**Meta-analytic functional decoding.** To provide a functional characterization for the white matter clusters derived from AF parcellation, we employed a custom meta-analytic approach, specifically designed to account both for the discrete nature of binary clusters and the white matter nature of the underlying spatial maps. Meta-analytic decoding was based on the Neuroquery database, which contains predictive activation maps estimated from over 7547 neuroscience terms[30].

Initially, we generated voxel-wise inverse distance maps from each cluster centroid to transform binary cluster maps into a continuous distribution of values reflecting the extent of each voxel belonging to each cluster. This process was carried out separately for the left and right AF clusters. Within voxel-wise distance maps, the value of each voxel indicates its proximity to cluster centroids, with higher values signifying closer distances. In an initial screening of terms from the Neuroquery database, these inverse distance maps underwent a masking procedure using a mean GM mask. Subsequently, the masked maps were utilized as inputs for the Neuroquery image search tool (https://github.com/neuroquery/neuroquery_image_search), retrieving the top 20 most correlated terms to each cluster distance map, resulting in a total of 120 neuroscience terms (2 hemispheres × 3 clusters × 20 terms). To this initial selection, a first screening procedure was applied by discarding duplicate terms and terms referring to anatomy or localization (e.g., "left", "right", "cortex", "hemisphere" "pfc"). A template tractogram was obtained for left and right AF by summing all the subject-specific AF tractograms in standard space. Subsequently, the unthresholded statistical maps corresponding to each of the remaining terms were retrieved and converted to track-weighted predictive term maps using the left and right template tractograms via MRtrix3's *tckmap* command employing the option -scalar_map to provide the statistical maps as input[34,35].

Finally, pairwise Pearson's correlation was employed to quantify the similarity between each cluster distance map and the resulting track-weighted term maps. To address the spatial autocorrelation (SA) properties of arcuate maps, statistical significance was assessed using a permutational approach described in ref. 36. This approach involved the generation of SA-preserving surrogate maps through 1000 permutations. The resulting *p* values underwent correction for multiple comparisons using the Benjamini-Hochberg method. The effect sizes of correlations were assessed by computing the $R^2$ determination coefficient.

### Reporting summary
Further information on research design is available in the Nature Portfolio Reporting Summary linked to this article.

## Data availability
The primary dataset (HCP) was provided by the Human Connectome Project, WU-Minn Consortium (Principal Investigators: David Van Essen and Kamil Ugurbil; 1U54MH091657). The data are openly available from https://www.humanconnectome.org/ The "Leipzig Study for Mind-Body-Emotion Interactions" (LEMON) data used as a validation dataset was provided by the Mind-Body-Emotion group at the Max Planck Institute for Human Cognitive and Brain Sciences. The data are openly available from https://www.nitrc.org/projects/mpilmbb. The maps obtained in the present work are available at https://github.com/BrainMappingLab.

## Code availability
The codes employed in the present work are available at https://github.com/BrainMappingLab.

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

## Acknowledgements

The primary dataset (HCP) was provided by the Human Connectome Project, WU-Minn Consortium (Principal Investigators: David Van Essen and Kamil Ugurbil; 1U54MH091657), founded by the 16 NIH institutes and centers that support the NIH Blueprint for Neuroscience Research, and by the McDonnell Center for Systems Neuroscience at Washington University. We also gratefully acknowledge the Mind-Body-Emotion group at the Max Planck Institute for Human Cognitive and Brain Sciences, for the data of the "Leipzig Study for Mind-Body-Emotion Interactions" (LEMON) that have been used as validation dataset. This research was funded by the Italian Ministry of Health, Current Research Funds 2023. This work was also supported by DOD N° W81XWH-19-1-0810 (A.Q., D.M., and Alb.C). Alb.C is also supported by NEXTGENERATIONEU (NGEU) funded by the Ministry of University and Research (MUR), National Recovery and Resilience Plan (NRRP), Mission 4, Component 2, project MNESYS (Project Code 0000006, CUP D93C22000930002)—A multiscale integrated approach to the study of the nervous system in health and disease (Concession Decree No. 1553 of 11.10.2022). This project received funding from the Donders Mohrmann Fellowship No. 2401512 (SJF, NEUROVARIABILITY). MTdS is supported by HORIZON- INFRA-2022 SERV (Grant No. 101147319) "EBRAINS 2.0: A Research Infrastructure to Advance Neuroscience and Brain Health", by the European Union's Horizon 2020 research and innovation program under the European Research Council (ERC) Consolidator grant agreement No. 818521 (DISCONNECTOME), the University of Bordeaux's IdEx "Investments for the Future" program RRI "IMPACT", and the IHU "Precision & Global Vascular Brain Health Institute—VBHI" funded by the France 2030 initiative (ANR-23-IAHU-0001).

## Author contributions

G.A.B. conceived the study, implemented the methods, performed the analyses, and wrote the manuscript. V.N. implemented part of the methods and revised the manuscript. A.Q. revised the manuscript and provided funding. A.G. wrote and revised the manuscript. A.I. implemented part of the methods and performed part of the analyses. Ant.C. wrote and revised the manuscript. D.M. provided fundamental intellectual comments and revised the manuscript. M.A. provided fundamental intellectual and methodological comments and extensively revised the manuscript. S.J.F. provided fundamental intellectual and methodological comments and extensively revised the manuscript. M.T.d.S. provided fundamental intellectual and methodological comments and extensively revised the manuscript. Alb.C. conceived and coordinated the study, implemented the methods, performed the analyses, wrote and revised the manuscript, and provided funding.

## Competing interests

The authors have nothing to declare. MTdS is an Editorial Board Member for *Communications Biology*, but was not involved in the editorial review of, nor the decision to publish this article.
