## [Transparent Peer Review file · Communications Biology]

Functional anatomy and topographical organization of the frontotemporal arcuate fasciculus

Corresponding Author: Professor Alberto Cacciola

Version 0:

Reviewer comments:

Reviewer #1

(Remarks to the Author)

This manuscript uses rank-weighted dynamic functional connectivity to investigate the functional anatomy of the arcuate fasciculus. The manuscript is well written and methodologically sound. I appreciate the replication of the findings in a second dataset of lesser quality. These kinds of approaches are urgently needed in the neuroimaging space to address the replication issue. Below are several minor points which could strengthen the manuscript.

The abstract is missing important information: the fact that the study used 2 datasets, one with test-retest data, what the basic demographic info of participants was, and what the implications of the findings are.

The last paragraph of the introduction already discusses some of the results and implications of the paper. This information should be preserved for later sections. Instead, the final paragraph of the introduction should culminate in a formulation of hypotheses based on the reviewed literature.

Minor: please specify number of b0s acquired in reverse phase-encoded direction.

It might be interesting to present Figure 1 with the remaining components, even though they were not the winning component models. It is reassuring to see that $k=2$ seems to be consistent across time-points and between very different datasets, but they only cover a very small number of fibres in the arcuate. It would be interesting to see which areas the remaining components connect.

Reviewer #2

(Remarks to the Author)

1. Brief Summary of the Manuscript

This paper explores whether distinct cortical origins and termination patterns within the frontotemporal arcuate fasciculus (AF) correspond to specific language functions. The authors employed track-weighted dynamic functional connectivity (TW-Dfc) to study the AF's structure and function in healthy participants. Their research identified three distinct clusters within the AF—ventral, middle, and dorsal frontotemporal AF—each linked to different frontal and temporal termination regions and associated with various language production and comprehension abilities.

2. Overall Impression of the Work

The finding of three distinct clusters within the AF that correspond to various language abilities is novel and will be of significant interest to the research community. The writing and structure of the paper are well-organized, and the methods and statistical analyses employed are both innovative and appropriate. While the paper is strong overall, some minor revisions could further enhance its clarity and impact.

3. Specific Comments, with Recommendations for Addressing Each Comment

Introduction

Comment 1: The introduction briefly discusses the three segments of the AF but the focus of the research is solely on the fronto-temporal segment. It is not clear why the researchers chose to focus exclusively on this segment.

Recommendation 1: Clarify why the study focuses on the fronto-temporal segment and not the anterior or posterior segments. In the last paragraph of the introduction, make it explicit that the focus is on the fronto-temporal segment and include the aims and hypotheses of the study, removing any findings from this section.

Methods

Comment 2: In section 1: Participants and data acquisition, age and gender are reported. Additional information in this section would be beneficial.

Recommendation 2: Do you have information on additional demographics such as years of education and handedness? Were left-handed participants removed from the study such as in previous studies investigating the arcuate fasciculus by Catani et al (2005) and Kenney et al (2017) (DOI: 10.1515/tnsci-2017-0018)? Could you report any statistical differences in gender and age within and between primary and test-retest datasets?

Comment 3: Different preprocessing pipelines were used for the primary and test-retest datasets. The authors decided to keep the preprocessing pipelines different as in their previous work. Some additional information about potential limitations of this approach would be helpful.

Recommendation 3: It would be useful to address some of the potential limitations of using different preprocessing pipelines in Section 4 (Technical Issues and Limitations).

Comment 4: In Section 2.4, it is not explicitly stated that Tractseg was used to segment the fronto-temporal AF.

Recommendation 4: In Section 2.4, clearly state that the focus of segmentation is on the fronto-temporal AF to avoid assumptions that the indirect tracts were also segmented.

Recommendation 5: In Figure 4B, it would be beneficial to include labels on the diagram to clearly identify the three clusters and their distinct cortical terminations.

Results

Comment 6: There is a discrepancy between the text and figure regarding the colours of clusters.

Recommendation 6: Ensure that the colours mentioned in the text (line 190) align with the figure.

Discussion

Recommendation 7: In line 417, consider adding Kenney et al. (2017) (DOI: 10.1515/tnsci-2017-0018) as an additional reference. Their study reports potential right hemisphere compensatory strategies for language function in individuals with psychosis, which would support the discussion on the right AF's role in language function recovery and compensatory strategies

Conclusion

Recommendation 8: Remove any mention of hypotheses from the conclusion section and ensure they are properly stated in the introduction.

Reviewer #3

(Remarks to the Author)

Thank you for inviting me to review this submission to Communications Biology titled "Functional Anatomy and topographical organization of the frontotemporal arcuate fasciculus". Here are some comments and suggestions for the authors:

Brief summary of the manuscript:

- This is an interesting study analyzing the functional connectivity of the AF, describing the topographical organization and dividing it according to different connections utilizing processing techniques of DTI in healthy subjects.

Overall impression of the work:

- This radiological study adds insightful analysis of the topographical and functional distribution of the AF. The methods performed are adequate and their proposition is interesting. Validation of this study is necessary to confirm the results acquired in their experiments.

Specific comments, with recommendations for addressing each comment

- The introduction sections is so extensive that it seems like a review itself. Please revise and reduce it to 2 or 3 short paragraphs.

- All preprocessing and processing descriptions are adequate for DWI, DTI/tractography, and BOLD. Detailed step-by-step processing is described and referenced. However, it's too long and makes it uneasy to follow. I'd recommend you summarize it as much as possible, only for the most remarkable aspects, and let the rest of the information in Supplementary material.

- Please describe in the methods section the statistical analysis (what was considered significant, tests used, etc).

- Clusters are reasonable to group bundles and separate functional analysis. Results are interesting and comparisons between bundles as well as right/left AF are remarkable.

- In Figure 4 there are some neurological functions that are not readable. I'd suggest putting only the most remarkable ones and listing the others in the figure legend or throughout the text only. Otherwise, plots and other representations are sufficient to provide information for the 3 new subdivisions of the AF.
- Definitely, these results should be confirmed in future studies acquiring topographic and functional information from cadaveric specimens and intraoperative awake cortical and subcortical stimulation in non-health subjects (the gold standard for adequate brain mapping). Otherwise, these are only approximations of potential functions using predictions with Neuroquery. This must be discussed.
- ICA is adequate for analysis and was performed adequately.
- Complementary connectivity matrices between both anterior and posterior ends of the AF would help to understand the data presented in this study.
- These results suggest not only that parcellation and differentiation of different bundles of AF is necessary and evident using this processing, but also provide information to confirm and continue studying the meta-networking theory of cerebral function. This should be also discussed (check DOI: 10.1152/physrev.00033.2019)
- If resources for data analysis are from other studies then it's unnecessary to provide their information about the IRV/ethics approvals, they are already referenced.
- The last paragraph of the introduction is presented as conclusions, rather than the main objective of the study. Please revise and improve as needed.
- Please revise that some references are not adequately formatted throughout the manuscript. (Check line 66)
- For some reason, I cannot access the information embedded in the "Links to the publications related to the meta-analytic terms" file. Please revise and provide it.
- 144 references are out of expectations for a non-review article. Double-check if those are necessary, it's likely that when summarizing both the introduction and methods section would be sufficient.

Version 1:

Reviewer comments:

Reviewer #1

(Remarks to the Author)

The authors have addressed all of my comments.

Reviewer #2

(Remarks to the Author)

I have carefully reviewed the changes and am pleased to confirm that I fully accept them. I am satisfied with the final version and am happy to proceed with publication.

Reviewer #3

(Remarks to the Author)

Thank you for addressing all my suggestions and comments. I feel this new version of your manuscript has been improved significantly.

Point-by-point reply to Reviewers' comments

Dear Editor and Reviewers,

we would like to thank You for the valuable comments on our paper, which has been extensively revised in the present R1 form.

To simplify your review, all the major modifications generated to address your comments are reported in **red** characters in the new text of the draft.

Reviewers' comments:

Reviewer #1 (Remarks to the Author):

This manuscript uses rank-weighted dynamic functional connectivity to investigate the functional anatomy of the arcuate fasciculus. The manuscript is well written and methodologically sounds. I appreciate the replication of the findings in a second dataset of lesser quality. These kinds of approaches are urgently needed in the neuroimaging space to address the replication issue. Below are several minor points which could strengthen the manuscript.

REPLY: We thank the Reviewer for their positive feedback.

The abstract is missing important information: the fact that the study used 2 datasets, one with test-retest data, what the basic demographic info of participants was, and what the implications of the findings are.

REPLY: Thank you for identifying the omission. We completed the description of missing information in the abstract:

<< We used rank-weighted dynamic functional connectivity, a hybrid imaging technique, to study the AF structure and function in two distinct datasets of healthy subjects. >>

and

REPLY: Our findings may have relevant implications for the understanding of the functional anatomy of the AF as well as its contribution to linguistic and non-linguistic functions. >>

The last paragraph of the introduction already discusses some of the results and implications of the paper. This information should be preserved for later sections. Instead, the final paragraph of the introduction should culminate in a formulation of hypotheses based on the reviewed literature.

REPLY: We agree with the Reviewer. We have now revised and improved the final paragraph of the introduction accordingly.

<< In the present work, we adapt these methods to characterize the functional anatomy of the human frontotemporal AF, building on the hypothesis that independent branches within the AF may be segregated by their distinct activity profiles. We obtained bundle-specific tw-dFC time series of the AF by combining high-quality resting-state and diffusion data. Using a hard clustering approach based on independent component analysis (ICA), we aimed at identifying anatomically and functionally dissociable AF clusters in an unsupervised, data-driven fashion, according to dynamic changes in functional connectivity at the streamline endpoints²⁹. Finally, we peered into the functional meaning of such anatomical organization by applying a meta-analytic decoding approach based on the NeuroQuery predictive model and database¹. >>

Minor: please specify number of b0s acquired in reverse phase-encoded direction.

REPLY: We have now specified the number of b0 volumes acquired in reverse phase-encoding direction. As we have moved the entire preprocessing section details to supplementary material according to Reviewer #3's suggestion, this information can be now retrieved in Supplementary Methods 1.1., for the primary, test-retest datasets (HCP):

<< DWI volumes were acquired with 90 directions per shell *in addition to 18 non-diffusion-weighted ($b = 0 \text{ mm/s}^2$) volumes*, and a spatial isotropic resolution of 1.25 mm^2 . >>

and in Supplementary Methods 1.2. for the validation dataset (LEMON):

<< DWI data (single shell, $b = 1000 \text{ s/mm}^2$) were acquired using a multi-band accelerated sequence with spatial isotropic resolution = 1.7 mm , and 60 diffusion-encoding directions *plus 7 non-diffusion-weighted ($b = 0 \text{ s/mm}^2$) volumes*. >>

It might be interesting to present Figure 1 with the remaining components, even though they were not the winning component models. It is reassuring to the that $k=2$ seems to be consistent across time-points and between very different datasets, but they only cover a very small number of fibres in the arcuate. It would be interesting to see which areas the remaining components connect.

REPLY: Thank you for your feedback. We are unsure if we fully understand your comment. If you are referring to the last panel of Figure 1, which depicts the methodological frameworks, please note that only a few streamlines belonging to the AF have been highlighted with different colors for illustration purposes. The coloured streamlines traverse two different black squares (schematically representing two voxels), and the average functional connectivity value at the endpoints of the streamlines traversing those voxels has been computed. The coloured streamlines therefore do not represent the entire $k=2$ clusters, but rather ideally represent the reconstruction of tw-dFC time series. We hope this clarifies the representation in the figure. Should there be specific aspects requiring further clarification, we are open to addressing them.

Reviewer #2 (Remarks to the Author):

1. Brief Summary of the Manuscript

This paper explores whether distinct cortical origins and termination patterns within the frontotemporal arcuate fasciculus (AF) correspond to specific language functions. The authors employed track-weighted dynamic functional connectivity (TW-Dfc) to study the AF's structure and function in healthy participants. Their research identified three distinct clusters within the AF—ventral, middle, and dorsal frontotemporal AF—each linked to different frontal and temporal termination regions and associated with various language production and comprehension abilities.

2. Overall Impression of the Work

The finding of three distinct clusters within the AF that correspond to various language abilities is novel and will be of significant interest to the research community. The writing and structure of the paper are well-organized, and the methods and statistical analyses employed are both innovative and appropriate. While the paper is strong overall, some minor revisions could further enhance its clarity and impact.

REPLY: We thank the Reviewer for appreciating our manuscript and their insightful comments which helped us to improve the overall quality of our work.

3. Specific Comments, with Recommendations for Addressing Each Comment

Introduction

Comment 1: The introduction briefly discusses the three segments of the AF but the focus of the research is solely on the fronto-temporal segment. It is not clear why the researchers chose to focus exclusively on this segment.

Recommendation 1: Clarify why the study focuses on the fronto-temporal segment and not the anterior or posterior segments. In the last paragraph of the introduction, make it explicit that the focus is on the fronto-temporal segment and include the aims and hypotheses of the study, removing any findings from this section.

REPLY: We thank the reviewer for the constructive contribution. We decided to focus on the fronto-temporal segment of the arcuate fasciculus instead of the anterior and posterior segments, as we found substantial agreement in the field that these anatomically distinct parts of the arcuate fasciculus may subservise distinct cognitive functions in the context of language processing; on the other hand, the existence of independent sub-units within the frontotemporal AF is still questioned both from an anatomical and functional perspective. We have now clarified this point and improved the Introduction section accordingly as follows:

<< Anatomically, the AF has been subdivided into several segments: the long (frontotemporal or arcuate proper), anterior (frontoparietal, largely equivalent to the third branch of the superior longitudinal fasciculus, SLF3), and posterior (temporoparietal) segments^{3,4}. Converging evidence from works combining tractography and functional MRI, or lesion mapping studies in clinical populations, suggests that these distinct anatomical segments may contribute at different levels in language-specific cognitive processes. The frontoparietal segment of the arcuate fasciculus has been associated with phonology-to-movement mapping and phonology-based word retrieval^{5,6}, the parietotemporal segment to reading and word comprehension^{7,8}, and the long segment to low-level phonemic and phonological processing in the context of language production, including word and non-words (sublexical) repetition^{9,10}. While there is consensus that frontoparietal and parietotemporal segments represent distinct anatomical and functional units within the AF, the precise role of the direct, frontotemporal segment of the AF is still a matter of lively debate. A growing line of evidence suggests that the frontotemporal AF may take part in higher-order language production processes involving the integration of lexical and phonological information, such as naming and phonologic fluency¹¹⁻¹⁴. It has been proposed that this functional dissociation between

phonological and semantic processing in the frontotemporal AF may be grounded in the underlying bundle anatomy, as suggested by ex vivo and in vivo anatomical findings of “ventral” and “dorsal” frontotemporal sub-segments, with distinct course, origin, and termination^{15,16}. It has been hypothesized that the ventral segment would mediate mostly phonemical-phonological processing, while the dorsal segment would be involved in lexical-semantic processes^{17,18}. >>

and

<< In the present work, we adapt these methods to characterize the functional anatomy of the human frontotemporal AF, building on the hypothesis that independent branches within the AF may be segregated by their distinct activity profiles. We obtained bundle-specific tw-dFC time series of the AF by combining high-quality resting-state and diffusion data. Using a hard clustering approach based on independent component analysis (ICA), we aimed at identifying anatomically and functionally dissociable AF clusters in an unsupervised, data-driven fashion, according to dynamic changes in functional connectivity at the streamline endpoints²⁹. >>

Methods

Comment 2: In section 1: Participants and data acquisition, age and gender are reported. Additional information in this section would be beneficial.

Recommendation 2: Do you have information on additional demographics such as years of education and handedness? Were left-handed participants removed from the study such as in previous studies investigating the arcuate fasciculus by Catani et al (2005) and Kenney et al (2017) (DOI: 10.1515/tnsci-2017-0018)? Could you report any statistical differences in gender and age within and between primary and test-retest datasets?

REPLY: We thank the reviewer for having raised this point. Notably, it was not possible to statistically assess differences in age due to the differing age ranges between the two datasets. However, the first dataset (HCP-young adults)¹⁹ consists exclusively of young adults aged 25-35, while the second dataset (LEMON) combines a subsample of young adults (aged 20-35) with a subsample of healthy older adults (aged 59-77)²⁰.

Regarding left-handed participants, they have not been excluded as representing less than 10% of both samples (9.04% in the HCP dataset; 9.81 in the LEMON dataset). Education was not directly compared between samples as it has been collected with different measures between the two datasets: the HCP dataset measures the education years directly while the LEMON dataset employs the last achieved qualification, referring to the German educational system. Gender differences are statistically significant as the LEMON dataset has a significantly higher proportion of female participants ($\chi^2=18.14$, $p<0.001$). Additional demographics, including information about handedness, have been added to the section Methods Section 1 (Data acquisition and preprocessing):

<< Specifically, two distinct subsamples of patients from the HCP collection have been employed for the present work: the first subsample (primary dataset) consisted of 210 healthy participants (males=92, females=118, age range 22-36 years), and the second (test-retest dataset) included 44 participants with available test-retest MRI scans (males = 13; females = 31; age range: 22–36 years).³⁸ The validation dataset includes 3T structural, diffusion, and rs-fMRI data of 213 healthy subjects (males=138, females=75, age range 20-70 years) from the Leipzig Study for Mind-Body-Emotion Interactions (LEMON) dataset (http://fcon_1000.projects.nitrc.org/indi/retro/MPI_LEMON.html)²⁰. Left-handed participants have not been excluded from the analysis as representing less than 10% in both samples (9.04% in the HCP dataset; 9.81% in the LEMON dataset).

The two datasets come with remarkable differences in demographics, data acquisition and preprocessing, that are described in full details in Supplementary Methods, as well as in the relevant reference publications^{19–22}. Notably, it was not possible to statistically assess differences in age due to the differing age ranges between the two datasets. However, the first dataset (HCP-young adults)

consists exclusively of young adults aged 25-35, while the second dataset combines a subsample of young adults (aged 20-35) with a subsample of healthy older adults (aged 59-77). Gender differences are significant across the two datasets as the LEMON dataset has a significantly higher proportion of female participants ($\chi^2=18.14, p <0.001$). >>

Comment 3: Different preprocessing pipelines were used for the primary and test-retest datasets. The authors decided to keep the preprocessing pipelines different as in their previous work. Some additional information about potential limitations of this approach would be helpful.

Recommendation 3: It would be useful to address some of the potential limitations of using different preprocessing pipelines in Section 4 (Technical Issues and Limitations).

REPLY: We thank the Reviewer for bringing up this important point. Potential limitations of using different preprocessing pipelines between the primary and validation datasets have been briefly discussed in Discussion Section 4 (Technical Issues and Limitations):

<< Some limitations of the current approach need to be acknowledged. First, the validation dataset does not perfectly match the primary dataset, showing relevant differences in several experimental conditions. Such differences include demographical (larger age range, different gender proportion) as well as technical factors affecting both DWI (single shell, low b-value, no filtering) and rs-fMRI (lower temporal resolution, different denoising pipeline). While we chose to maintain the differences between datasets to highlight the reliability of our findings further, this approach makes it impossible to quantify the extent to which differences observed between the two datasets are driven by differences in the experimental pipeline rather than genuine, inter-individual anatomical and functional variability. >>

Comment 4: In Section 2.4, it is not explicitly stated that Tractseg was used to segment the fronto-temporal AF.

Recommendation 4: In Section 2.4, clearly state that the focus of segmentation is on the fronto-temporal AF to avoid assumptions that the indirect tracts were also segmented.

REPLY: We have modified the new Methods Section 2 (Bundle-specific tractography and tw-dFC) accordingly:

<< For bundle-specific tractography of the frontotemporal AF, TractSeg (<https://github.com/MIC-DKFZ/TractSeg/>), a convolutional neural network-based tract segmentation approach, was employed. >>

and

<< Participant-level binary tract masks and ending masks obtained from TractSeg were employed to guide tractography of the *frontotemporal AF*, ... >>

Recommendation 5: In Figure 4B, it would be beneficial to include labels on the diagram to clearly identify the three clusters and their distinct cortical terminations.

REPLY: We thank the Reviewer for suggesting this improvement to Figure 4B, which has been now modified accordingly by adding labels for the distinct cortical terminations of the three clusters. The new figure has also been reported below.

Results

Comment 6: There is a discrepancy between the text and figure regarding the colours of clusters.

Recommendation 6: Ensure that the colours mentioned in the text (line 190) align with the figure.

REPLY: Thank you for identifying the discrepancy. The figure legend has been modified by reporting the proper colours.

<< **Figure 4. Clustering of component spatial maps reveals a tripartite organization of the AF. A)** Scatter plots of component values in the component space and the resulting silhouette plots. For each voxel, weights of component 1 and component 2 are plotted in the resulting component space; colors progress from blue to red across component 1 and from blue to green across component 2. **B)** Ventral, middle, and dorsal clusters of the AF and their meta-analytic characterization. Clusters are both rendered in 3D volumetric space and projected on the cortical surface to emphasize the cortical termination patterns. Labels indicate gyral structures on the left and sulcal structures on the right. *MFG: middle frontal gyrus; IFG: inferior frontal gyrus; STG: superior temporal gyrus; MTG: middle*

temporal gyrus; ITG: inferior temporal gyrus; MFS: middle frontal sulcus; IFS: inferior frontal sulcus; FOP: frontal operculum; POP; parietal operculum; TOP; temporal operculum; STS: superior temporal sulcus; MTS: middle temporal sulcus. Word clouds are derived from correlation values obtained from meta-analytic decoding. Cyan: ventral cluster; orange: middle cluster; magenta: dorsal cluster. C) Violin plots of component weights across the three clusters. lh: left hemisphere; rh: right hemisphere. >>

Discussion

Recommendation 7: In line 417, consider adding Kenney et al. (2017) (DOI: 10.1515/tnsci-2017-0018) as an additional reference. Their study reports potential right hemisphere compensatory strategies for language function in individuals with psychosis, which would support the discussion on the right AF's role in language function recovery and compensatory strategies

REPLY: The suggested reference has been added.

Conclusion

Recommendation 8: Remove any mention of hypotheses from the conclusion section and ensure they are properly stated in the introduction.

REPLY: The conclusion section has been rephrased properly.

Reviewer #3 (Remarks to the Author):

Thank you for inviting me to review this submission to Communications Biology titled “Functional Anatomy and topographical organization of the frontotemporal arcuate fasciculus”. Here are some comments and suggestions for the authors:

Brief summary of the manuscript:

- This is an interesting study analyzing the functional connectivity of the AF, describing the topographical organization and dividing it according to different connections utilizing processing techniques of DTI in healthy subjects.

Overall impression of the work:

- This radiological study adds insightful analysis of the topographical and functional distribution of the AF. The methods performed are adequate and their proposition is interesting. Validation of this study is necessary to confirm the results acquired in their experiments.

REPLY: We thank the Reviewer for appreciating our manuscript.

Specific comments, with recommendations for addressing each comment

- The introduction sections is so extensive that it seems like a review itself. Please revise and reduce it to 2 or 3 short paragraphs.

REPLY: The Introduction section has now been reduced by approximately 50%, while keeping the most relevant information to understand the rationale of the current study.

<< The arcuate fasciculus (AF) is a prominent association pathway in the human brain. Since its first description in the 19th century, this white matter bundle has been considered crucial for language processing^{23–26}, and for other relevant cognitive functions²⁶. Accordingly, this structure has been extensively investigated in the last decades, using post-mortem dissection methods and, more recently, in vivo diffusion-weighted tractography^{5–8}.

Anatomically, the AF has been subdivided into several segments: the long (frontotemporal or arcuate proper), anterior (frontoparietal, largely equivalent to the third branch of the superior longitudinal fasciculus, SLF3), and posterior (temporoparietal) segments^{3,4}. Converging evidence from works combining tractography and functional MRI, or lesion mapping studies in clinical populations, suggests that these distinct anatomical segments may contribute at different levels in language-specific cognitive processes. The frontoparietal segment of the arcuate fasciculus has been associated with phonology-to-movement mapping and phonology-based word retrieval^{5,6}, the parietotemporal segment to reading and word comprehension^{7,8}, and the long segment to low-level phonemic and phonological processing in the context of language production, including word and non-words (sublexical) repetition^{9,10}. While there is consensus that frontoparietal and parietotemporal segments represent distinct anatomical and functional units within the AF, the precise role of the direct, frontotemporal segment of the AF is still a matter of lively debate. A growing line of evidence suggests that the frontotemporal AF may take part in higher-order language production processes involving the integration of lexical and phonological information, such as naming and phonologic fluency^{11–14}. It has been proposed that this functional dissociation between phonological and semantic processing in the frontotemporal AF may be grounded in the underlying bundle anatomy, as suggested by ex vivo and in vivo anatomical findings of “ventral” and “dorsal” frontotemporal sub-segments, with distinct course, origin, and termination^{15,16}. It has been hypothesized that the ventral segment would mediate mostly phonemical-phonological processing, while the dorsal segment would be involved in lexical-semantic processes^{17,18}. >>

- All preprocessing and processing descriptions are adequate for DWI, DTI/tractography, and BOLD. Detailed step-by-step processing is described and referenced. However, it’s too long and makes it

uneasy to follow. I'd recommend you summarize it as much as possible, only for the most remarkable aspects, and let the rest of the information in Supplementary material.

REPLY: We have reduced the methods section by keeping only the most remarkable aspects to make it easier to read. Additional information on the complete description of the datasets, preprocessing and processing steps have been reported in Supplementary material.

- Please describe in the methods section the statistical analysis (what was considered significant, tests used, etc).

REPLY: Detailed description of the statistical calculations employed in the meta-analytic decoding is described in section 3.4:

<< Finally, pairwise Pearson's correlation was employed to quantify the similarity between each cluster distance map and the resulting track-weighted term maps. To address the spatial autocorrelation (SA) properties of arcuate maps, statistical significance was assessed using a permutational approach described in Burt et al.²⁷. This approach involved the generation of SA-preserving surrogated maps through 1000 permutations. The resulting p-values underwent correction for multiple comparisons using the Benjamini-Hochberg method. The effect sizes of correlations were assessed by computing the R² determination coefficient". >>

- Clusters are reasonable to group bundles and separate functional analysis. Results are interesting and comparisons between bundles as well as right/left AF are remarkable.

REPLY: We thank the Reviewer for the comment.

- In Figure 4 there are some neurological functions that are not readable. I'd suggest putting only the most remarkable ones and listing the others in the figure legend or throughout the text only. Otherwise, plots and other representations are sufficient to provide information for the 3 new subdivisions of the AF.

REPLY: We thank the Reviewer for this observation. The word clouds in Figure 4 are designed to provide the reader with an immediate understanding of the most representative results obtained from the meta-analytic decoding. The readability of words/functions in the word cloud corresponds to their correlation values: larger, more readable words indicate stronger correlations, while smaller, less readable words represent weaker correlations. This visual distinction helps to quickly convey the hierarchy of neurological functions related to the three new subdivisions of the AF. For this reason, we would prefer keeping Figure 4 as in its current form. To enhance clarity, results of the meta-analytic decoding are also provided Tables 1-2, ensuring that detailed information is accessible without compromising the visual impact of the figure.

- The results should be confirmed in future studies acquiring topographic and functional information from cadaveric specimens and intraoperative awake cortical and subcortical stimulation in non-health subjects (the gold standard for adequate brain mapping). Otherwise, these are only approximations of potential functions using predictions with Neuroquery. This must be discussed.

REPLY: We agree with the Reviewer on this important aspect, which has been now reported at the end of the Discussion section 3.

<< In synthesis, while comparing the results of our functional and anatomical parcellation of the AF at rest to meta-analytic estimates derived from task-based functional MRI does not provide conclusive evidence on the involvement of specific tract segments, our work allows to draw hypotheses on the functional meaning of specific anatomical segments of the AF. The confirmation of these findings is therefore left to further investigations, through intra-operative electrode recording during awake surgery¹². >>

- ICA is adequate for analysis and was performed adequately.

REPLY: We thank the Reviewer for the comment.

- Complementary connectivity matrices between both anterior and posterior ends of the AF would help to understand the data presented in this study.

REPLY: We appreciate the Reviewer's suggestion to consider the anterior (frontoparietal or SLF3) and posterior (temporoparietal) segments in addition to the frontotemporal segment of the AF. While we acknowledge that including these segments could provide a more comprehensive understanding of the AF's complex functional anatomy, we chose to focus on the arcuate proper (frontotemporal segment) for the following reasons (see also reply to R1), which have now been added to the manuscript:

'Add new section'

- These results suggest not only that parcellation and differentiation of different bundles of AF is necessary and evident using this processing, but also provide information to confirm and continue studying the meta-networking theory of cerebral function. This should be also discussed (check DOI: 10.1152/physrev.00033.2019)

REPLY: We thank the Reviewer for pointing out this important topic. The meta-networking theory of cerebral function has been now considered in the discussion section:

<< The arcuate fasciculus (AF) is a prominent association pathway in the human brain. Since its first description in the 19th century, this white matter bundle has been considered crucial for language processing^{23–26}, and for other relevant cognitive functions²⁶. Accordingly, this structure has been extensively investigated in the last decades, using post-mortem dissection methods and, more recently, in vivo diffusion-weighted tractography^{5–8}.

Anatomically, the AF has been subdivided into several segments: the long (frontotemporal or arcuate proper), anterior (frontoparietal, largely equivalent to the third branch of the superior longitudinal fasciculus, SLF3), and posterior (temporoparietal) segments^{3,4}. Converging evidence from works combining tractography and functional MRI, or lesion mapping studies in clinical populations, suggests that these distinct anatomical segments may contribute at different levels in language-specific cognitive processes. The frontoparietal segment of the arcuate fasciculus has been associated with phonology-to-movement mapping and phonology-based word retrieval^{5,6}, the parietotemporal segment to reading and word comprehension^{7,8}, and the long segment to low-level phonemic and phonological processing in the context of language production, including word and non-words (sublexical) repetition^{9,10}. While there is consensus that frontoparietal and parietotemporal segments represent distinct anatomical and functional units within the AF, the precise role of the direct, frontotemporal segment of the AF is still a matter of lively debate. A growing line of evidence suggests that the frontotemporal AF may take part in higher-order language production processes involving the integration of lexical and phonological information, such as naming and phonologic fluency^{11–14}. It has been proposed that this functional dissociation between phonological and semantic processing in the frontotemporal AF may be grounded in the underlying bundle anatomy, as suggested by ex vivo and in vivo anatomical findings of “ventral” and “dorsal” frontotemporal sub-segments, with distinct course, origin, and termination^{15,16}. It has been hypothesized that the ventral segment would mediate mostly phonemical-phonological processing, while the dorsal segment would be involved in lexical-semantic processes^{17,18}. >>

- If resources for data analysis are from other studies then it's unnecessary to provide their information about the IRV/ethics approvals, they are already referenced.

REPLY: We have deleted the sections related to the IRB/ethics approvals.

- The last paragraph of the introduction is presented as conclusions, rather than the main objective of the study. Please revise and improve as needed.

REPLY: We thank the Reviewer for the helpful suggestion. The last paragraph of the introduction has been improved accordingly as also suggested by R1:

<< *In the present work, we adapt these methods to characterize the functional anatomy of the human frontotemporal AF, building on the hypothesis that independent branches within the AF may be segregated by their distinct activity profiles. We obtained bundle-specific tw-dFC time series of the AF by combining high-quality resting-state and diffusion data. Using a hard clustering approach based on independent component analysis (ICA), we aimed at identifying anatomically and functionally dissociable AF clusters in an unsupervised, data-driven fashion, according to dynamic changes in functional connectivity at the streamline endpoints*²⁹. Finally, we peered into the functional meaning of such anatomical organization by applying a meta-analytic decoding approach based on the NeuroQuery predictive model and database¹. >>

- Please revise that some references are not adequately formatted throughout the manuscript. (Check line 66)

REPLY: We have carefully revised and adequately formatted the references throughout the manuscript.

- For some reason, I cannot access the information embedded in the “Links to the publications related to the meta-analytic terms” file. Please revise and provide it.

REPLY: Thank you for highlighting this issue. The links to the publications related to the meta-analytic terms have been included in the Supplementary Information .pdf file and they should be now easily accessible.

- 144 references are out of expectations for a non-review article. Double-check if those are necessary, it's likely that when summarizing both the introduction and methods section would be sufficient.

REPLY: The number of references has been minimized as much as possible.

Yours sincerely,

Alberto Cacciola & Co-Authors

References

1. Dockès, J. *et al.* NeuroQuery, comprehensive meta-analysis of human brain mapping. *Elife* **9**, (2020).
2. Sotiropoulos, S. N. *et al.* Advances in diffusion MRI acquisition and processing in the Human Connectome Project. *Neuroimage* **80**, 125–43 (2013).
3. Catani, M., Jones, D. K. & ffytche, D. H. Perisylvian language networks of the human brain. *Ann Neurol* **57**, 8–16 (2005).
4. Mandonnet, E., Sarubbo, S. & Petit, L. The Nomenclature of Human White Matter Association Pathways: Proposal for a Systematic Taxonomic Anatomical Classification. *Front Neuroanat* **12**, (2018).
5. Schwartz, M. F., Faseyitan, O., Kim, J. & Coslett, H. B. The dorsal stream contribution to phonological retrieval in object naming. *Brain* **135**, 3799–3814 (2012).
6. Bohland, J. W., Bullock, D. & Guenther, F. H. Neural Representations and Mechanisms for the Performance of Simple Speech Sequences. *J Cogn Neurosci* **22**, 1504–1529 (2010).
7. Thiebaut de Schotten, M., Cohen, L., Amemiya, E., Braga, L. W. & Dehaene, S. Learning to Read Improves the Structure of the Arcuate Fasciculus. *Cerebral Cortex* **24**, 989–995 (2014).

8. Turken, A. U. & Dronkers, N. F. The Neural Architecture of the Language Comprehension Network: Converging Evidence from Lesion and Connectivity Analyses. *Frontiers in System Neuroscience* **5**, (2011).
9. Shinoura, N. *et al.* Damage to the left ventral, arcuate fasciculus and superior longitudinal fasciculus -related pathways induces deficits in object naming, phonological language function and writing, respectively. *International Journal of Neuroscience* **123**, 494–502 (2013).
10. Breier, J. I., Hasan, K. M., Zhang, W., Men, D. & Papanicolaou, A. C. Language Dysfunction After Stroke and Damage to White Matter Tracts Evaluated Using Diffusion Tensor Imaging. *American Journal of Neuroradiology* **29**, 483–487 (2008).
11. Sarubbo, S. *et al.* Structural and functional integration between dorsal and ventral language streams as revealed by blunt dissection and direct electrical stimulation. *Hum Brain Mapp* **37**, 3858–3872 (2016).
12. Sarubbo, S. *et al.* Mapping critical cortical hubs and white matter pathways by direct electrical stimulation: an original functional atlas of the human brain. *Neuroimage* **205**, 116237 (2020).
13. Giampiccolo, D. & Duffau, H. Controversy over the temporal cortical terminations of the left arcuate fasciculus: a reappraisal. *Brain* **145**, 1242–1256 (2022).
14. Duffau, H. *et al.* New insights into the anatomo-functional connectivity of the semantic system: a study using cortico-subcortical electrostimulations. *Brain* **128**, 797–810 (2005).
15. Yagmurlu, K., Middlebrooks, E. H., Tanriover, N. & Rhoton, A. L. Fiber tracts of the dorsal language stream in the human brain. *J Neurosurg* **124**, 1396–1405 (2016).
16. Fernández-Miranda, J. C. *et al.* Asymmetry, connectivity, and segmentation of the arcuate fascicle in the human brain. *Brain Struct Funct* **220**, 1665–1680 (2015).
17. Glasser, M. F. & Rilling, J. K. DTI Tractography of the Human Brain’s Language Pathways. *Cerebral Cortex* **18**, 2471–2482 (2008).
18. Janssen, N. *et al.* Dissociating the functional roles of arcuate fasciculus subtracts in speech production. *Cerebral Cortex* **33**, 2539–2547 (2023).
19. Van Essen, D. C. *et al.* The Human Connectome Project: A data acquisition perspective. *Neuroimage* **62**, 2222–2231 (2012).
20. Babayan, A. *et al.* A mind-brain-body dataset of MRI, EEG, cognition, emotion, and peripheral physiology in young and old adults. *Sci Data* **6**, 180308 (2019).
21. Glasser, M. F. *et al.* The minimal preprocessing pipelines for the Human Connectome Project. *Neuroimage* **80**, 105–124 (2013).
22. Mendes, N. *et al.* A functional connectome phenotyping dataset including cognitive state and personality measures. *Sci Data* **6**, 180307 (2019).
23. Negwer, C. *et al.* Loss of Subcortical Language Pathways Correlates with Surgery-Related Aphasia in Patients with Brain Tumor: An Investigation via Repetitive Navigated Transcranial Magnetic Stimulation–Based Diffusion Tensor Imaging Fiber Tracking. *World Neurosurg* **111**, e806–e818 (2018).
24. Fridriksson, J. *et al.* Anatomy of aphasia revisited. *Brain* **141**, 848–862 (2018).
25. Price, C. J. The anatomy of language : contributions from functional neuroimaging. *J. Anat* **197**, 335–359 (2000).
26. Forkel, S. J. *et al.* Anatomical evidence of an indirect pathway for word repetition. *Neurology* **94**, e594–e606 (2020).
27. Burt, J. B., Helmer, M., Shinn, M., Anticevic, A. & Murray, J. D. Generative modeling of brain maps with spatial autocorrelation. *Neuroimage* **220**, 117038 (2020).